# Comparative and Phylogenetic Analysis of Complete Chloroplast Genomes of Five *Mangifera* Species

**DOI:** 10.3390/genes16060666

**Published:** 2025-05-30

**Authors:** Yujuan Tang, Xiangyan Yang, Shixing Luo, Guodi Huang, Yu Zhang, Ying Zhao, Riwang Li, Limei Guo, Mengyang Ran, Aiping Gao, Jianfeng Huang

**Affiliations:** 1Guangxi Subtropical Crops Research Institute, Nanning 530001, China; gxtyj6722@126.com (Y.T.); luoshixing595@163.com (S.L.); gxrzszy0624@163.com (Y.Z.); zhaoying-222@163.com (Y.Z.); gxlrw@126.com (R.L.); 13260269320@163.com (L.G.); r1612668128@outlook.com (M.R.); 2Guangxi Engineering Research Center of Green and Efficient Development for Mango Industry, Nanning 530001, China; 3Guangxi Key Laboratory of Quality and Safety Control for Subtropical Fruits, Nanning 530001, China; 4Chinese Academy of Tropical Agricultural Sciences, Haikou 571101, China; aipinggao@126.com (A.G.); huangjian1984xy@163.com (J.H.)

**Keywords:** *Mangifera*, chloroplast genome, Illumina, polymorphism analysis, phylogenetic analysis

## Abstract

**Background/Objectives**: Mango, which is known as the “King of Tropical Fruits”, is an evergreen plant belonging to the Anacardiaceae family. It belongs to the genus *Mangifera*, which comprises 69 species of plants found in tropical and subtropical regions, including India, Indonesia, the Malay Peninsula, Thailand, and South China. However, research on the structural information of complete chloroplast genomes of *Mangifera* is limited. **Methods**: The rapid advancement of high-throughput sequencing technology enables the acquisition of the entire chloroplast (cp) genome sequence, providing a molecular foundation for phylogenetic research. This work sequenced the chloroplast genomes of six *Mangifera* samples, performed a comparative analysis of the cp genomes, and investigated the evolutionary relationships within the *Mangifera* genus. **Results**: All six *Mangifera* samples showed a single circular molecule with a quadripartite structure, ranging from 157,604 bp to 158,889 bp in length. The number of RNA editing sites ranged from 60 to 61, with ndhB exhibiting the highest number of RNA editing sites across all species. Seven genes—namely, *atpB*, *cemA*, *clpP*, *ndhD*, *petB*, *petD*, and *ycf15*—exhibited a Ka/Ks value > 1, suggesting they may be under positive selection. Phylogenetic analysis revealed that *Mangifera siamensis* showed a close relationship between *Mangifera indica* and *Mangifera sylvatica*. **Conclusions**: Our comprehensive analysis of the whole cp genomes of the five *Mangifera* species offers significant insights regarding their phylogenetic reconstruction. Moreover, it elucidates the evolutionary processes of the cp genome within the *Mangifera* genus.

## 1. Introduction

The mango is an evergreen species in the genus *Mangifera* of the Anacardiaceae family. This genus comprises 69 species found globally, predominantly in tropical and subtropical regions such as India, Indonesia, Thailand, and South China [1,2]. The leaves exhibit two colors: fuchsia for new growth and green for mature foliage, and it is a much favored tropical fruit [3,4]. It is widely planted because of its high economic and nutrient value [5]. Due to its unique taste and pleasing appearance, it is widely loved by fruit consumers and is known as the “King of Tropical Fruits” [5]. Southeast Asian countries have a history of mango cultivation spanning millennia [6]; the fruit spread to Africa and South America centuries ago, and they have recently developed several cultivated species suitable for their local climate [5,7]. Among these species, *M. indica* is the predominant one grown for commercial fruit production.

Phylogenetic study of *Mangifera* species has garnered significant attention in research [8,9]. Recent investigations have utilized chloroplast genome markers [10,11] and nuclear gene sequences [12,13] to explore evolutionary relationships within this genus. However, the findings remain inconsistent, and many analyses have struggled to achieve fully resolved evolutionary trees. A key challenge stems from the predominantly outcrossing nature of cultivated and wild mangoes, as they are generally self-incompatible. As a result, hybridization is likely when different species grow in close proximity. Taxonomic ambiguities further compound these challenges, as exemplified by *M. odorata* and *M. siamensis*, whose phylogenetic positions remain contested due to overlapping morphological traits [14], while others, such as *M. sylvatica*, are hypothesized as wild progenitors of cultivated mango but lack genomic validation. Therefore, elucidating the molecular evolutionary links within the genus *Mangifera* with higher resolution is essential to enhance the effective use of wild relatives in forthcoming breeding initiatives.

Chloroplasts are essential metabolic centers in terrestrial plants and algae, facilitating photosynthesis and producing amino acids, nucleotides, fatty acids, and other metabolites crucial for their physiology and growth [15]. The maternal inheritance, haploid condition, and restricted recombination of the chloroplast genome augment its applicability for phylogenetic and genomic studies across plant groups [16,17,18,19]. Beyond phylogenetic inference, chloroplast genome analyses facilitate comprehensive investigations into structural genomic divergence, including large-scale rearrangements such as inverted repeat (IR) boundary shifts and gene order variations that delineate lineage-specific evolutionary trajectories [20]. These studies further enable the quantification of molecular evolutionary dynamics through metrics such as codon usage bias and nucleotide substitution patterns, which collectively reveal adaptive constraints across coding and non-coding regions [21]. Simultaneously, chloroplast microsatellites (cpSSRs) provide high-resolution markers for probing intraspecific genetic diversity and population structure, while systematic profiling of RNA editing sites uncovers post-transcriptional regulatory mechanisms critical for plastid gene expression fidelity [22]. Such multilayered genomic examinations advance both our fundamental understanding of plant adaptation and practical applications in biodiversity conservation. Niu et al. used whole cp genomes of five *Mangifera* species to investigate the evolutionary relationships within the genus. The information is limited due to the small number of taxa with published cp genome sequences [23]. More recently, sequences of chloroplast genomes and a specific collection of common single-copy genes from the nuclear genome of 14 *Mangifera* species were compared to examine evolutionary connections within the genus [24]. However, the study of comparative analyses between cp genomes, such as SSR, Ka/Ks, codon analysis, and RNA editing, remains insufficiently explored.

In this study, we sequenced the cp genomes of six *Mangifera* samples, including *M. siamensis* for the first time. We conducted a cp genome-based analysis and provided a comprehensive account of the cp genome assembly, annotations, and simple sequence repeats (SSRs). We performed a phylogenetic analysis of *Mangifera* using whole chloroplast genome sequences from newly sequenced species in conjunction with a previously reported one (*M. longipes*). The aims were to (i) define the structure of the *Mangifera* cp genome, (ii) find regions of the *Mangifera* cp genome that are highly divergent and suitable for DNA barcoding, and (iii) determine the evolutionary relationships within the *Mangifera* genus.

## 2. Materials and Methods

### 2.1. Plant Material

We collected leaves from six samples representing five species of the genus *Mangifera*. Fresh leaves of the plants were collected and immediately stored at −80 °C. Samples were selected to maximize geographic representation across major distribution zones of *M. indica* and related species. Of the six samples, three were collected from Guangxi, China: *M. indica*-OK104092 in Hezhou, *M. persiciforma*-OK104090 in Baise, and *M. sylvatica*-OK104091 in Nanning. One species was collected in Yunnan, China (*M. siamensis*-MZ926796 in Xishuangbanna); one species was from Thailand (*M. indica*-OK104089 in Chiengmai); and another was from the United States (*M. odorata*-MZ926795 in Florida). The two *M. indica* samples were used for comparing intra-species variation across the Asian cultivation. All plant species were cultivated under standardized management practices at the Guangxi Subtropical Crops Research Institute (Nanning, China), ensuring consistent growth conditions.

### 2.2. DNA Extraction, Sequencing, and Annotation

Total genomic DNA was extracted using the modified CTAB method [25]. Sample integrity and purity were verified with agarose gel electrophoresis and quantified using a NanoDrop 2000 spectrophotometer (Thermo Fisher Scientific, Waltham, MA, USA). Qualified genomic DNA was processed with the NexteraXT DNA Library Preparation Kit (Illumina, San Diego, CA, USA) to generate 350 bp fragment libraries, followed by paired-end sequencing on the Illumina NovaSeq 6000 platform (Illumina). Each sample generated about 11.4 million PE150 reads (paired-end 150bp), which were subjected to quality control using the NGS QC Tool Kit v2.3.3 [26]. GFA files created by SPAdes 3.11.0 software [27] facilitated the verification of circular assembly via graph visualization. Annotation of these chloroplast sequences was performed using Plastid Genome Annotator (https://github.com/quxiaojian/PGA, accessed on 11 May 2023) [28], and BLAST search from Genbank was used to evaluate the results.

### 2.3. Codon Usage and RNA Editing Analysis

The sequence of *Mangifera* longipes (MN917210) was downloaded from the NCBI for comparative analyses. We selected all the PCGs in each of the six samples using Unipro UGENES v.36.0 [29]. Then, codon usage frequency in each of these samples was analyzed using codonW v. 1.4.2 (http://codonw.sourceforge.net/, accessed on 23 May 2023). Synonymous codon usage and relative synonymous codon usage (RSCU) were obtained to determine whether the plastid genes were under selection. Then, we used the PREP-cp (Predictive RNA Editors for Plants chloroplast) [30] to identify the RNA editing sites by comparing the PCGs in the six samples, and the cut-off score was set to C = 0.8 to find all true RNA editing sites.

### 2.4. Identification of Repeat Sequences and Simple Sequence Repeats (SSR)

SSR identification was detected on the chloroplast genome sequences using the MIcroSAtellite identification tool with threshold parameters set as follows: mononucleotides (≥8 repeats), dinucleotides (≥5), tri- to hexanucleotides (≥4/3/3/3 respectively), and a minimum 100 bp spacing between adjacent SSRs [31]. Dispersed (forward, reverse, palindrome, and complementary) repeats were determined by running the REPuter program with a minimum repeat size of 30 bp and similarities of 90%. Tandem repeats were identified by running the web-based Tandem Repeats Finder, with alignment parameters being set to 2, 7, and 7 for matches, mismatches, and indels, respectively [32].

### 2.5. Sequence Variation Map and Variations in Inverted Repeat (IR) Sequences

The cp genome sequences were aligned with MAFFT v7.4.29 [33], and the complete chloroplast genome of MI098 was employed as a reference and compared with the chloroplast genomes of others using the mVISTA program in the Shuffle-LAGAN mode [34]. Then, we used IRscope to analyze IR Region Contraction (https://irscope.shinyapps.io/irapp/, accessed on 26 May 2023).

### 2.6. Pi and Ka/Ks

A sliding window analysis was performed to find rapidly developing molecular markers for nucleotide variability (Pi) in the whole chloroplast genome using DnaSp v6.12.03 [35]. The window length was set at 800 bp, with a 100 bp step size. We computed the Ka and Ks to assess selection pressure using the branch-site model in the program codeml of the PAML v4.9 package. A neutral branch-site model (Mod-el  =  2, NSsites  =  2, Fix  =  1, and Fix ω  =  1) was applied.

### 2.7. Phylogenetic Analysis of Chloroplast Genomes

A phylogenomic study was performed using six newly sequenced cp genomes from *Mangifera* samples, in conjunction with eight *Mangifera* species obtained from GenBank, with Citrus aurantifolia serving as the outgroup for the phylogenetic assessment. The sequence of PCGs of *Mangifera* samples was extracted using Unipro UGENES v.36.0 [28]. The phylogenetic analysis workflow initiates with multiple sequence alignment using MAFFT, where unaligned FASTA files are processed through the software’s default auto-mode to generate aligned sequence outputs. Then, the validation of alignment quality was conducted via the visualization tool AliView to ensure proper positional homology and minimal aberrant gaps. Subsequently, phylogenetic tree construction is performed using FastTree, where protein sequences employ the Le Gascuel (LG) evolutionary model with γ correction, while nucleotide sequences utilize the GTR model with γ correction. Branch reliability is assessed through bootstrap analysis (1000 replicates), and the final Newick format tree files, containing branch lengths and support values, are generated. Finally, tree visualization and annotation are conducted using FigTree, which enables customization of branch esthetics and explicit labeling of support metrics.

## 3. Results and Discussion

### 3.1. Genomic Characteristics of Chloroplast

The preliminary dataset of six *Mangifera* specimens was meticulously filtered to eliminate adapters and low-quality reads, yielding a substantial dataset of 3–5 Gb per species, with a minimum coverage depth of 100×. The complete circular chloroplast genomes with tetrad architecture were reconstructed through iterative assembly and splicing methodologies, containing characteristic quadripartite structures (Figure 1): a large single-copy (LSC) region, a small single-copy (SSC) region, and two inverted repeats (IRs). The structural composition of the complete chloroplast genomes in the six examined *Mangifera* samples adheres to the conventional configuration observed in most plants, characterized by a singular circular quadripartite molecule [36]. The entire chloroplast genomes of the samples ranged from 157,604 bp to 158,889 bp. The biggest species was *M. odorata*, whereas the smallest was *M. siamensis*, with a difference of 1285 bp. The size of *Mangifera* cp genomes studied here is comparable with previously reported *Mangifera* species [23,24]. The lengths of the IR region in the samples were very consistent, with just a 29 bp variation. *M. indica* TH and *M. indica* CN exhibited identical lengths of IR regions. In contrast, the length of LSC differed across species, with *M. siamensis* displaying the shortest size at 86,507 bp and *M. odorata* presenting the highest at 87,708 bp.

In terms of the number of genes across cp genomes, all five *Mangifera* species contained 37 tRNA genes. *M. indica* TH exhibited 89 protein-coding genes (PCGs), resulting in a total of 134 genes, while the other four species contained 88 PCGs, totaling 133 genes (Table 1). All species in this study had eight rRNAs in their chloroplast genomes. More genes have been annotated in our analysis compared to previous studies, which reported a total of 112 to 115 genes [24,37]. The GC concentration in these genomes is crucial for sequence stability. Elevated GC content is associated with greater DNA density, resulting in enhanced structural rigidity and reduced DNA flexibility [38]. The GC content of the seven chloroplast genomes covered around 37.82% to 37.90%, and they increased as the size of the chloroplast genomes decreased (Table 1).

### 3.2. Codon Usage

Although the universal codon table is conserved throughout all creatures, reflecting a similar heritage, biological evolution has resulted in the emergence of distinct biases. Diverse species exhibit preferences for certain synonymous codons, and within a single species, different proteins may preferentially use the same amino acid, a phenomenon referred to as codon bias. RSCU, a widely used measure for assessing codon bias, aids in alleviating the impact of amino acid composition linked to certain codons [21]. The PCGs in the full cp genome of Mangifera consist of 61 codons, corresponding to 20 amino acids. Overall, leucine was the most prevalent amino acid in all of the species, around 10.56–10.59%, covering about 2798 to 2804 codons. Cysteine was the least prevalent amino acid in the species, around 1.17–1.18%, covering 310 to 314 codons (Appendix A). RSCU values of more than 1 indicate significant codon bias, whereas values over 1.3 signal high-frequency codons [39]. Analysis of the Mangifera cp genome reveals 20 high-frequency codons, mostly ending with either A or U. Conversely, codons ending with a C-terminal, such as AGC (Ser), UAC (Tyr), GAC (Asp), and CGC (Arg), exhibit RSCU values lower than 0.45 (Figure 2 and Appendix A). A previous study indicated that, within the cp genome of angiosperms, the majority of codons exhibit a pronounced A/T bias in the third position [40], corroborating our findings. The primary cause of this situation may be associated with the prevalence of A or T in the infrared region [41].

### 3.3. RNA Editing Analysis

RNA editing serves as a critical regulatory mechanism for gene expression in angiosperm chloroplast (cp) genomes, with its functional significance spanning RNA maturation and adaptive responses [22,42]. RNA editing analyses were conducted using cp genomes of newly sequenced species in conjunction with a previously reported one (*M. longipes*). This study identified between 60 and 61 RNA editing sites, totaling 422 among seven samples. Two *M. indica*, *M. siamensis*, *M. odorata*, and *M. sylvatica* exhibited identical RNA editing sites, totaling 60 each. *M. persiciforma* and *M. longipes* showed 61 RNA editing sites, including 1 more site each for *ccsA*, *rpoB*, and *matK*, but displaying 2 fewer sites for *rpoC2* compared to the other five samples (Figure 3). The gene *ndhB* had the most RNA editing sites in all species, followed by *ndhD*. None of the 422 RNA editing sites found in our analysis were situated in the third position (Appendix A). It is worth noting that the absence of anticipated RNA editing sites in inactive regions is likely attributable to the limitations of the PREP-Mt prediction methodology rather than the nonexistence of RNA editing in these locations. PREP-Mt provides only gene-level annotations, limiting fine-scale evolutionary analyses; the selection criteria used in PREP-Mt may be inadequate for identifying the modified variant [43]. Future studies integrating codon resolution prediction frameworks (e.g., MitoZ) or direct RNA-seq validation could resolve these methodological constraints, enabling precise mapping of RNA editing dynamics and their evolutionary significance across both coding and non-coding cp genome regions. All RNA editing occurred in converting C to T and in no other conversions, and 56.7% of these RNA editing sites occurred in conversions to leucine (Appendix A).

### 3.4. Non-Synonymous Substitutions (Ka) and Synonymous Substitutions (Ks) Analysis

Determining the Ka and Ks is essential for phylogenetic reconstruction and comprehending the evolutionary dynamics of PCGs in closely related species [44]. The Ka/Ks ratio serves as a metric to evaluate selection pressure on a specific PCG throughout evolution. A Ka/Ks ratio exceeding 1 signifies positive selection, a ratio equal to 1 denotes neutral selection, and a ratio below 1 indicates negative selection [45]. Here, we calculated the Ka/Ks of 35 PCGs across seven samples in a pairwise manner. Overall, the majority of genes underwent negative selection during evolution, shown by the fact that 28 out of 35 PCGs had Ka/Ks ratios below 1 across all comparisons. Seven genes, *atpB, cemA, clpP, ndhD, petB, petD*, and *ycf15*, exhibited a Ka/Ks value > 1. Comparable results have been documented across other plant species, indicating that genes such as *petB, petD, ndhD*, and *cemA* are subject to positive selection pressure throughout evolution [46,47,48]. The highest Ka/Ks ratio of 3.9142 was seen in the *petB* gene of *M. longipes* compared to other species, suggesting that this gene may have experienced positive selection throughout the development of *Mangifera* (Figure 4). The *petB* gene encodes the protein cytochrome b6, integral to the cytochrome b6f complex (Cytb6f), a component of the photosynthetic system. Within the electron transport chain, Cytb6f serves as an electron conduit between photosystem II (PSII) and photosystem I (PSI) [49], thereby playing a crucial role in photosynthesis. Positive selection in petB likely reflects adaptive optimization of electron transport kinetics, improving energy conversion efficiency under environmental stressors. The candidate genes under positive selection warrant further functional validation through transgenic complementation assays in model plants, which could elucidate their adaptive roles in *Mangifera* diversification.

### 3.5. Detection of Chloroplast Repeat Sequences and SSRs

The cp genome demonstrates parthenogenetic features, with significant simple sequence repeats (SSRs) polymorphism observed within conspecific individuals [50]. These variations enable SSRs to function as robust molecular markers for developmental biology and taxonomic discrimination [51], while also facilitating applications in population genetics and evolutionary studies [52]. In the present study, we identified 65 to 75 SSRs in seven *Mangifera* samples (Figure 5A). The mononucleotides encompassed the majority of SSRs, accounting for around 60.87% to 67.57%, as previously shown in other angiosperm research [19,53]. Pentanucleotides were only detected in *M. indica* TH, *M. persiciforma*, and *M. indica* CN. In the mononucleotides, C/G only covered around 4.17–10.87% of SSRs in different species. Prior research [54,55] has established that A/T bases were the most frequently employed among all categories of SSRs. The diminished nitrogen content of A/T bases relative to G/C bases suggests that an increase in A/T bases may lead to base mutations that necessitate less energy [56]. Additionally, the SSRs were detected in 10 PCGs across all samples; however, *M. indica*, *M. siamensis*, and *M. sylvatica* exhibited an additional SSR in a tRNA gene (*trnK-UUU*) (Appendix A). At present, there is a paucity of published research on SSR markers within the *Mangifera* cp genome. SSR discovery in the *Mangifera* cp genome might provide a foundation for the development of molecular markers and the investigation of genetic diversity within the *Mangifera* genus.

Long repeat sequences, defined as repetitions exceeding 30 bp, have a role in cp genome rearrangement [57]. Our research investigated four classes of interspersed genomic repeats—complementary, forward, palindromic, and inverse repeats—in seven chloroplast genomesOnly forward and palindromic repeats were detected, with inverse and complementary repeats absent across all samples (Figure 5B). It has been reported that complementary repeats are consistently infrequent across all species [16], coinciding with our results. The number of palindromic repeats was substantially greater than that of forward repeats, covering more than 60% of total long repeats. The number of long repeats ranged from 10 (*M. odorata*) to 20 (*M. longipes*) pairs. Most of the repeats (93.8%) varied from 30 to 52 bp in length, and only one pair of repeats was more than 26,360 bp in each species, which comprised IR regions. More than 50% of the long repeats were located in LSC or SSC regions. The long repeats were identified in *rps16, rps19, ycf1, ycf2, ndhA, psaA, psaB*, and *petD* (Appendix A).

### 3.6. IR Contraction and Expansion

IR expansion and contraction stand out as the major processes generating change in cp genome size, having a considerable effect on species evolution [20]. Thus, a detailed comparative analysis of the junctions of the IRs and two SCs, including LSC/IRA (JLA), LSC/IRB (JLB), SSC/IRA (JSA), and SSC/IRB (JSB), was conducted along with placement of adjacent genes in the chloroplast genomes of seven *Mangifera* samples. The length of IR regions was less variable among species (Figure 6), which was in accordance with Niu’s results [23]. However, previous investigations have shown that the varying lengths found in cp genomes result from expansion and contraction events within the IR regions [52], which was not the case in our study. Certain lineages may evolve mechanisms to stabilize IR regions over time. The ancestors of mango underwent a whole genome duplication event approximately 33 million years ago, leading to gene redundancy and functional diversification. This redundancy may have reinforced the stability of critical genes and restricted subsequent structural variation through purifying selection [58].

Seven genes—*rpl22*, *rps19*, *rpl2*, *ycf1*, *ndhF*, *trnH*, and *psbA*—were detected at the junction of the SC and IRs (Figure 6). In *M. persiciforma*, *rps19* was entirely situated within the IR regions, positioned 204 bp from JLB; conversely, in other species, *rps19* was located closer at 181 bp from JLB within the IR regions. In *M. indica* TH, *M. indica* CN, and *M. sylvatica*, *ycf1* was entirely found in the IR regions, 3 bp from JSB, while others were 5 bp from JSB. In terms of another *ycf1* located on JSA, the gene located in SSC ranged from 4512 bp to 4529 bp. In *M. longipes*, *rps19* was closer to JLA than *rpl2*, with a distance of 3 bp. In contrast, in other species, *rpl2* was nearer to JLA, with distances ranging from 151 to 174 bp. For the gene *ndhF*, it spans JSB in all species, with distances to JSB of approximately 34–36 bp. The location of *trnH* is on LSC in all species, with distances to JLA ranging from 58 to 82 bp. Taken together, the *Mangifera* species were highly conserved, as indicated by similar IR/SC borders.

### 3.7. Sequence Divergence Analysis

Comparing differences in cp genome sequences across different taxa enables the effective identification of information-rich DNA regions and promotes the development of methodologies for species identification and the investigation of population diversification [17]. We further analyzed the differences in the chloroplast sequences of seven *Mangifera* samples using both mVISTA and DnaSP6. The results indicated that the chloroplast genome sequences of *Mangifera* had very high sequence similarities (Figure 7). Overall, the majority of variations occurred in intergenic areas, whereas the coding regions remained rather stable. *Ycf1* and *ycf2* exhibited higher variation in comparison with the other coding genes. In contrast, great differences were identified in some intergenic regions, such as *trnH-psbA*, *ycf4-ccmA*, and *ndhF-rpl32*.

The nucleotide variation (Pi) of the seven samples ranged from 0 to 0.011. The IR regions exhibited no variable regions with Pi > 0.005. Both the LSC and SSC regions were more divergent than the IR regions in the chloroplast genome of *Mangifera* species, which was consistent with previous research [18]. Additionally, a previous study has suggested a greater vulnerability to mutations in non-coding areas than in coding areas [59]. In accordance with the results of mVISTA, the three most variable regions (Pi > 0.010) were *trnH-psbA*, *ycf4-cemA*, and *ndhF-rpl32*, with Pi > 0.010 (Figure 8). *Ycf4-cemA* was also reported as a hot spot by a previous study analyzing cp genomes of five *Mangifera* species. Six other regions exhibiting great variability (0.01 > Pi > 0.006) were identified inside the LSC region, including IGS *trnK-UUU-rps16*, *rps16-trnQ-UUG*, *petN-psbM*, *trnG-GCC-psbZ*, and *rpl36-rps8* (Appendix A). The significant variety in these areas, particularly within the LSC and SSC zones, supplied diverse data for the creation of markers for the molecular categorization and phylogenetic study of *Mangifera* species.

### 3.8. Phylogenetic Analysis

We constructed a phylogenetic tree of six samples of *Mangifera* sequenced in this study and eight other *Mangifera* species, using Citrus aurantiifolia (KJ865401) as an outgroup based on its taxonomic position within Sapindales, with the sequences of complete cp genomes and protein-coding genes, respectively (Figure 9). All 14 *Mangifera* samples were completely supported (100 BS) as monophyletic, then formed two branches. The first branch consisted of two *M. indica* from distinct origins, the principal species grown for commercial fruit production, subsequently clustering with *M. sylvatica* and finally *M. siamensis*. Previous research has shown a significant evolutionary connection between *M. indica* and *M. sylvatica* using RFLP [10], ITS marker analysis [13], and comprehensive chloroplast genome sequencing [23]. *M. siamensis* is a wild mango species mostly located in Myanmar, and cross-pollination with *M. indica* may have contributed to the emergence of some species. This work sequenced the chloroplast genome of *M. siamensis* for the first time and elucidated its evolutionary relationships with other species in this genus. In the present study, *M. siamensis* exhibited a close relationship with *M. indica*, which could be supported by morphological characteristics. It has been reported that the only morphological distinction between the two species is the presence of solitary stamens and staminodes [60].

Another branch comprised 10 *Mangifera* samples. Specifically, *M. longipes* and the two *M. persiciforma* individuals form a sister group with strong bootstrap support (100 BS), diverging from a broader group that includes seven *Mangifera* species. *M. quadrifida* branched into a separate clade containing *M. caloneura*, *M. macrocarpa*, *M. pentandra*, and *M. foetida*, and was identified to be grouped with *M. odorata* (100 BS). According to the AFLP marker analysis, *M. odorata* is suggested to be a hybrid of *M. indica* and *M. foetida*, exhibiting a greater affinity for *M. foetida* than for *M. indica* [13]. The close phylogenetic affinity between *M. odorata* and *M. foetida* (100 BS) might also reflect historical hybridization events, as suggested by their overlapping distributions in Malesia and shared floral morphologies despite divergent fruit traits. The current findings further corroborate that *M. odorata* is more closely related to *M. foetida* than to *M. indica*, based on both the complete cp genome and the coding genes of cp genome phylogenies.

The phylogenetic trees reconstructed from chloroplast genome sequences and protein-coding genes exhibited consistent evolutionary relationships among species, differing solely in their bootstrap support (BS) values. Notably, the majority of phylogenetic trees based on protein-coding genes demonstrated lower BS values compared to those constructed using chloroplast genome sequences, which was consistent with a previous study [61]. Protein-coding genes often exhibit higher functional constraints due to selective pressures, leading to slower evolutionary rates and reduced informative sites for resolving phylogenetic relationships. In contrast, chloroplast genomes contain abundant non-coding regions and synonymous substitutions in coding regions, which accumulate neutral mutations at a moderate rate, providing stronger phylogenetic signals. While this study focused on six representative taxa, future investigations should expand sampling to underrepresented wild relatives to better resolve phylogenetic relationships. In addition, while chloroplast data resolved maternal lineages robustly, their uniparental inheritance limits the detection of hybridization signals, which could be complemented by nuclear genome analyses.

## 4. Conclusions

In this study, six complete cp genomes of *Mangifera* (two *M. indica* from China and Thailand, *M. persiciforma*, *M. sylvatica*, *M. siamensis*, and *M. odorata*) were sequenced, assembled, and comparatively analyzed. The cp genomes of these samples contained 86 to 88 protein-coding genes, 8 rRNA, and 36 to 37 tRNA, as well as 65 to 75 SSRs and 60 to 61 RNA editing sites. The *Mangifera* species exhibited significant conservation, as shown by the comparable IR/SC boundaries and high sequence similarities demonstrated with mVISTA. Three intergenic spacers—namely, *trnH-psbA, ycf4-cemA*, and *ndhF-rpl32*—showed considerable diversity (Pi > 0.010). A preliminary phylogenetic tree was constructed using the whole cp genomes and protein-coding regions of the cp genomes of 14 *Mangifera* samples. Our study shows for the first time that *M. siamensis* has a close connection with *M. indica* and *M. sylvatica*. This work enhances the repository of cp genome sequences, hence improving species identification and aiding phylogenetic analysis in future *Mangifera* research.

## Figures and Tables

**Figure 1 genes-16-00666-f001:**
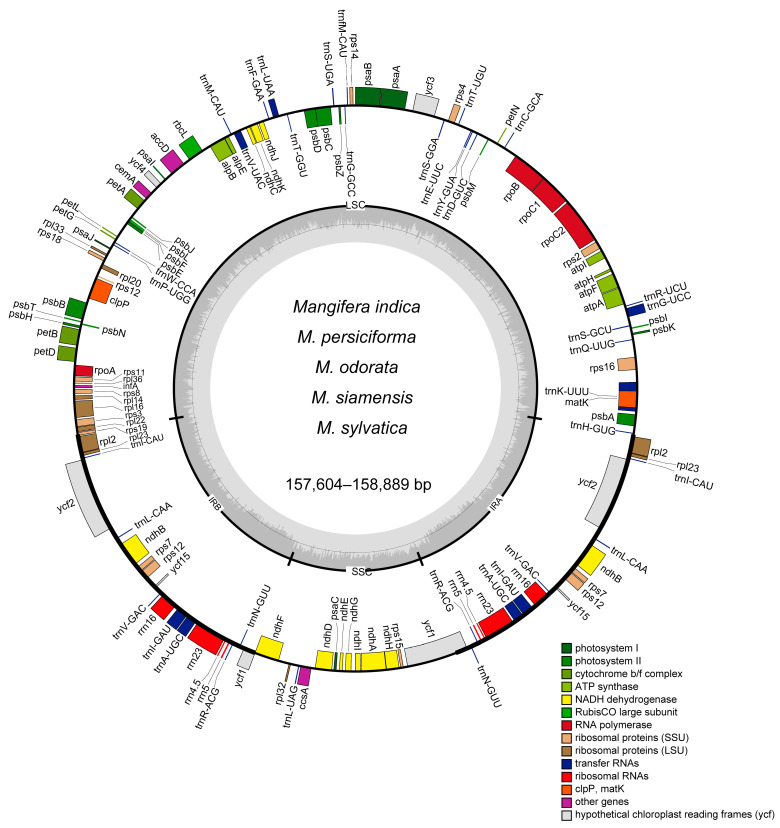
A gene map of the chloroplast genome for six Mangifera samples. Genes are transcribed in a clockwise direction on the outer circle and counterclockwise on the inner circle. Functional gene groups are color-coded for clarity. Variations in shading on the inner circle represent the GC content (darker gray) and AT content (lighter gray) of the chloroplast genome. SSC, small single copy. LSC, long single copy. IR, inverted repeat.

**Figure 2 genes-16-00666-f002:**
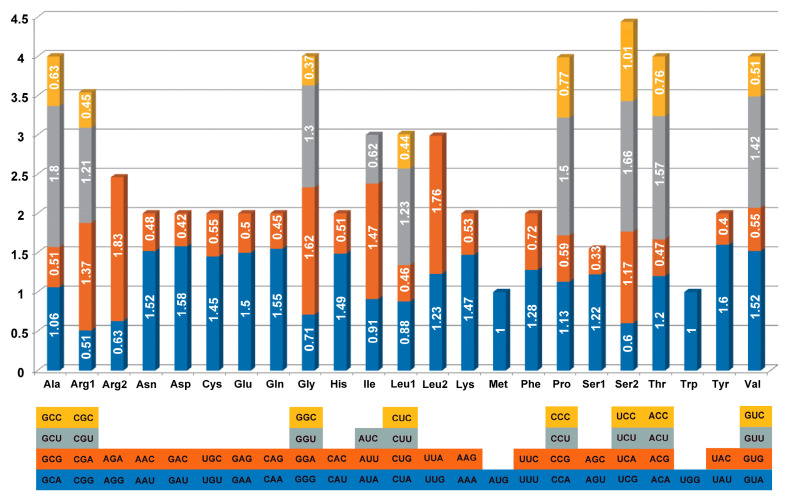
RSCU of chloroplast genes in Mangifera indica CH.

**Figure 3 genes-16-00666-f003:**
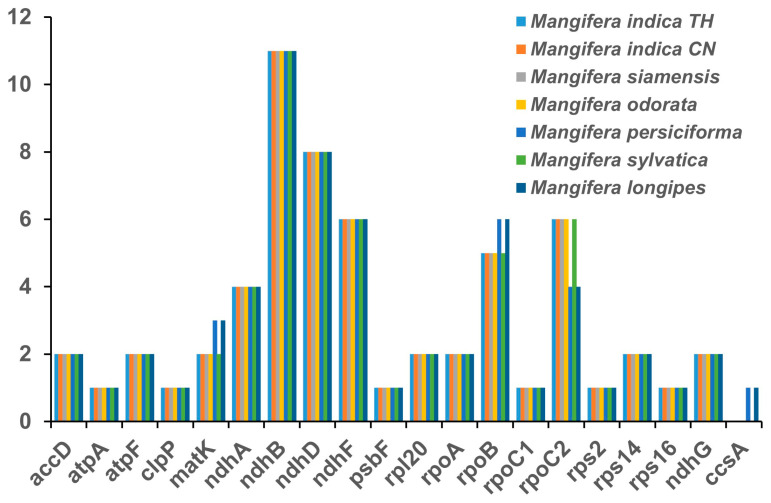
Number of RNA editing sites of chloroplast genes in seven *Mangifera* samples. The cp genome sequence of *M. longipes* (MN917210) was obtained from GenBank.

**Figure 4 genes-16-00666-f004:**
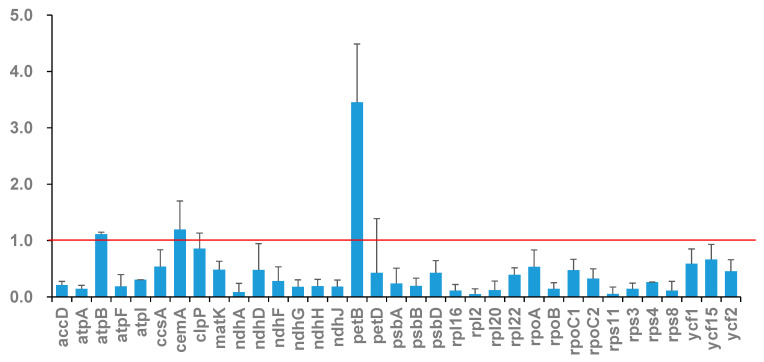
Value of Ka/Ks of chloroplast genes in seven *Mangifera* samples. The red dashed line indicates the neutral evolution threshold (Ka/Ks = 1).

**Figure 5 genes-16-00666-f005:**
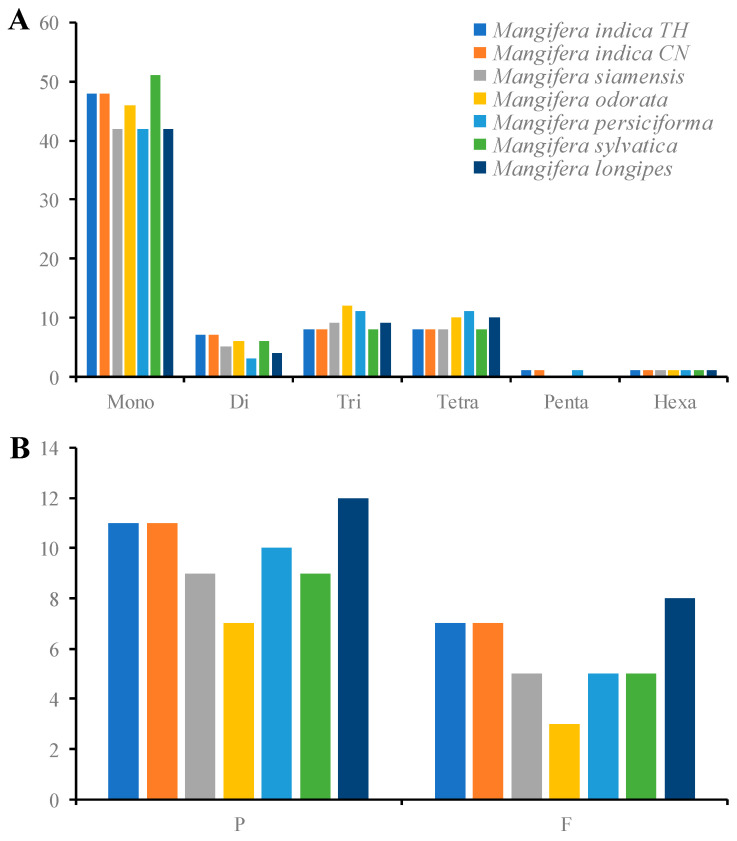
Analysis of simple sequence repeats (**A**) and long repeats (**B**) in the chloroplast genomes of seven *Mangifera* samples. P, palindromic repeats; F, forward repeats. The cp genome sequence of *M. longipes* (MN917210) was obtained from GenBank.

**Figure 6 genes-16-00666-f006:**
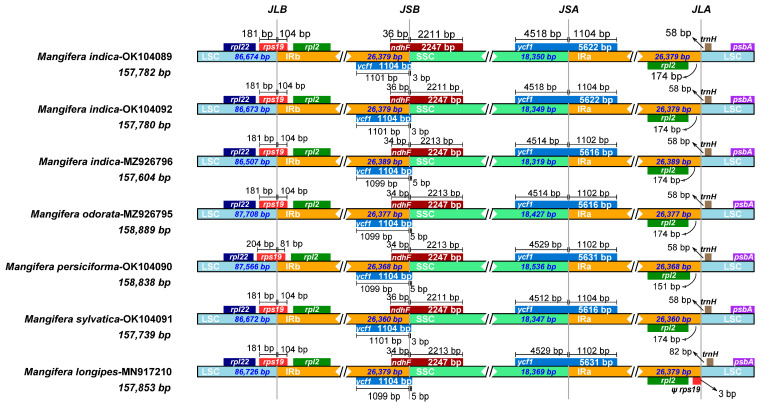
Comparison of junctions between four regions across chloroplast genomes of 7 *Mangifera* samples. JLB, the junction line between LSC and IRb; JSB, the junction line between IRb and SSC; JSA, the junction line between SSC and IRa; JLA, the junction line between IRa and LSC. Ψ, pseudogene.

**Figure 7 genes-16-00666-f007:**
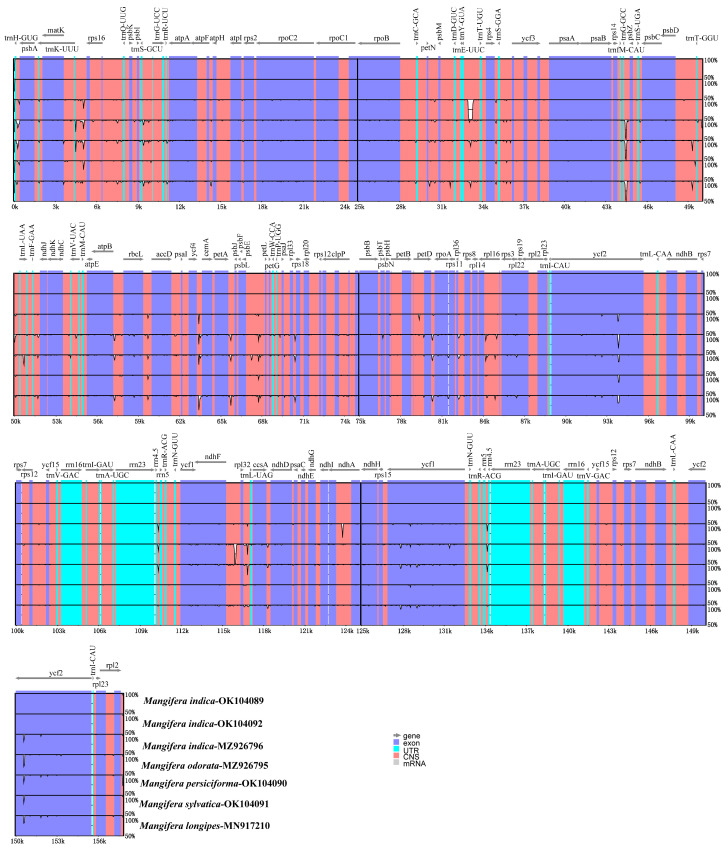
Global alignment of 7 chloroplast genomes of *Mangifera* using mVISTA. The *Y*-axis indicates the range of identity (50–100%). Alignment was performed using *M. indica* TH as a reference. Various colors differentiate between exons, introns, untranslated regions (UTRs), and conserved non-coding sequences (CNSs). The cp genome sequence of *M. longipes* (MN917210) was obtained from GenBank.

**Figure 8 genes-16-00666-f008:**
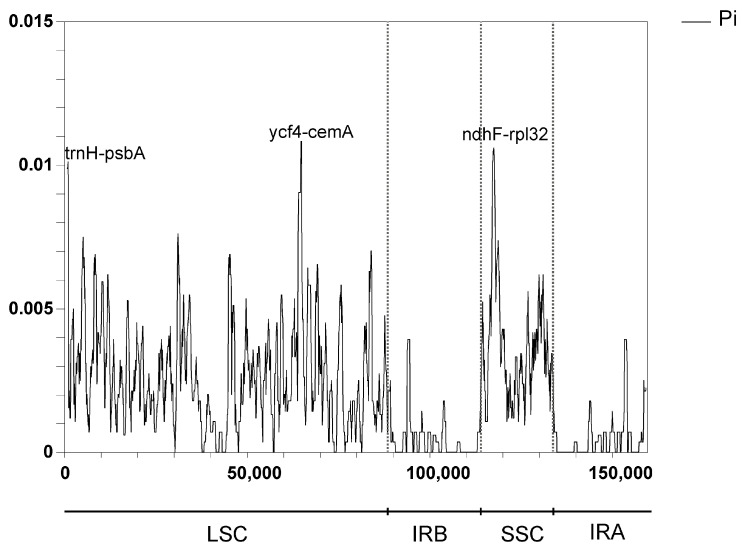
Nucleotide variability (Pi) values across chloroplast genomes of 7 *Mangifera* samples (window length: 600 bp; step size: 200 bp). The cp genome sequence of *M. longipes* (MN917210) was obtained from GenBank.

**Figure 9 genes-16-00666-f009:**
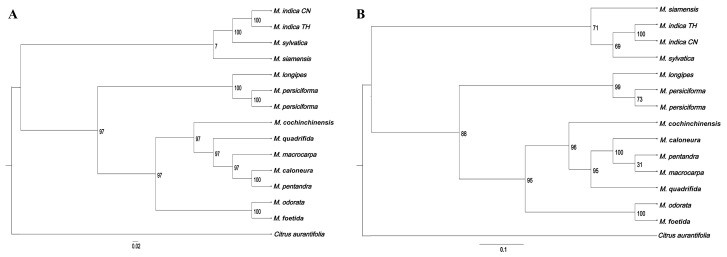
Preliminary phylogenetic tree of six *Mangifera* samples and their related species based on chloroplast genomes (**A**) and protein-coding genes of chloroplast genomes (**B**). Chloroplast genome sequences were downloaded from GenBank.

**Table 1 genes-16-00666-t001:** Features of six complete chloroplast genomes.

Species	*M. indica*	*M. indica*	*M. siamensis*	*M. odorata*	*M. persiciforma*	*M. sylvatica*
Accession number	OK104089	OK104092	MZ926796	MZ926795	OK104090	OK104091
Size	157,782	157,780	157,604	158,889	158,838	157,739
GC content	37.89%	37.89%	37.90%	37.81%	37.82%	37.89%
IR size	26,379	26,379	26,389	26,377	26,368	26,360
LSC size	86,674	86,673	86,507	87,708	87,566	86,672
SSC size	18,350	18,349	18,319	18,427	18,536	18,347
Genes	134	133	133	133	133	133
PCGs	89	88	88	88	88	88
tRNA	37	37	37	37	37	37
rRNA	8	8	8	8	8	8

## Data Availability

All data cited in this study are publicly available. The raw reads generated in this study were deposited in the NCBI database under BioProject PRJNA1264534 (SRA accessions: SRR33617711 to SRR33617716).

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
