# Peer review of "Comparative and Phylogenetic Analysis of Complete Chloroplast Genomes of Five Mangifera Species"

_genes, 2025, doi:10.3390/genes16060666_

Round 1
Reviewer 1 Report
Comments and Suggestions for Authors
This article presents interesting and up-to-date research on the chloroplast genomes of six samples from five Mangifera species, with the chloroplast genome sequence of Mangifera siamensis being reported for the first time. A comprehensive comparative analysis was conducted, including the structure of the chloroplast genomes, RNA-editing site analysis, identification of simple sequence repeats (SSRs), nucleotide variability, and molecular selection analysis (Ka/Ks). Based on the obtained data, phylogenetic analyses were performed, providing new insights into the evolutionary relationships within the Mangifera genus.
The manuscript is scientifically valuable, aligns well with current research trends in plant genomics, and has potential applications in species identification and future mango breeding programs. The article makes a significant contribution to the literature, especially by enriching the available chloroplast data for Mangifera, including for under-studied species.
However, the reviewed manuscript requires several important revisions, both editorial and methodological.
Major Points for Revision:
- English Language Quality:
The text contains numerous grammatical, stylistic, and punctuation errors. Examples include:
- “and it was widely planted by the reason of high economic and nutrient value” — incorrect structure.
- Frequent repetitions, poor sentence construction, and tense inconsistencies.
âž” Suggestion: A full professional language editing is necessary.
- Gaps in Methodology:
- Parameters used for data processing in Gblocks (e.g., block trimming settings) are not fully described.
- The sample selection strategy (especially the two indica individuals from different countries) should be justified.
- No detailed information is provided on sequencing depth (coverage), which is crucial for evaluating assembly quality.
- Positive Selection (Ka/Ks) Analysis:
- Merely stating Ka/Ks > 1 is insufficient to claim positive selection without appropriate statistical tests (e.g., FUBAR, MEME).
âž” Suggestion: Strengthen the selection analysis with statistical tests or explicitly discuss its limitations.
- Raw Data Availability:
- Authors state that the data are publicly available, but no accession numbers for raw reads (e.g., in the SRA database) are provided.
âž” Suggestion: Add accession numbers and direct links.
- RNA Editing Analysis:
- The use of PREP-Mt as the predictive tool should be justified, especially considering the availability of more updated alternatives.
- The limitations of the PREP-Mt method should be more extensively discussed.
- Phylogenetic Analysis:
- The biological significance of the obtained relationships should be more thoroughly discussed, including potential hybridization events (e.g., odorata vs M. foetida).
- The limitations of using only chloroplast DNA (absence of nuclear genome data) should be acknowledged.
- Figures and Tables:
- Some figures need more detailed legends, explaining abbreviations and symbols.
- On plots (e.g., Ka/Ks values), it would be beneficial to indicate thresholds (e.g., Ka/Ks = 1 line).
Questions for the Authors:
- What criteria were used in selecting the sampling locations?
- Are there additional data confirming the taxonomic identity of the studied individuals (e.g., voucher specimens)?
- Are there plans for further comparative studies on a larger number of wild Mangifera species?
- Is it possible to make the alignment files and phylogenetic trees used in the analyses publicly available?
- Do the authors intend to conduct functional validation of the genes identified under positive selection (e.g., petB, cemA)?
What Should Be Improved Before Publication:
- Professional English language editing.
- Full description of bioinformatics methods and sample selection criteria.
- Expanded positive selection analysis or clearly stating limitations.
- Provision of raw data accession numbers and links.
- Expanded interpretation of phylogenetic results, including potential hybridization scenarios.
The article is of high scientific value and certainly contributes new data to Mangifera chloroplast genomics research. However, especially concerning language quality, methodology, and data presentation.
Comments on the Quality of English Language
This article presents interesting and up-to-date research on the chloroplast genomes of six samples from five Mangifera species, with the chloroplast genome sequence of Mangifera siamensis being reported for the first time. A comprehensive comparative analysis was conducted, including the structure of the chloroplast genomes, RNA-editing site analysis, identification of simple sequence repeats (SSRs), nucleotide variability, and molecular selection analysis (Ka/Ks). Based on the obtained data, phylogenetic analyses were performed, providing new insights into the evolutionary relationships within the Mangifera genus.
The manuscript is scientifically valuable, aligns well with current research trends in plant genomics, and has potential applications in species identification and future mango breeding programs. The article makes a significant contribution to the literature, especially by enriching the available chloroplast data for Mangifera, including for under-studied species.
However, the reviewed manuscript requires several important revisions before publication, both editorial and methodological.
Major Points for Revision:
- English Language Quality:
The text contains numerous grammatical, stylistic, and punctuation errors. Examples include:
- “and it was widely planted by the reason of high economic and nutrient value” — incorrect structure.
- Frequent repetitions, poor sentence construction, and tense inconsistencies.
âž” Suggestion: A full professional language editing is necessary.
- Gaps in Methodology:
- Parameters used for data processing in Gblocks (e.g., block trimming settings) are not fully described.
- The sample selection strategy (especially the two indica individuals from different countries) should be justified.
- No detailed information is provided on sequencing depth (coverage), which is crucial for evaluating assembly quality.
- Positive Selection (Ka/Ks) Analysis:
- Merely stating Ka/Ks > 1 is insufficient to claim positive selection without appropriate statistical tests (e.g., FUBAR, MEME).
âž” Suggestion: Strengthen the selection analysis with statistical tests or explicitly discuss its limitations.
- Raw Data Availability:
- Authors state that the data are publicly available, but no accession numbers for raw reads (e.g., in the SRA database) are provided.
âž” Suggestion: Add accession numbers and direct links.
- RNA Editing Analysis:
- The use of PREP-Mt as the predictive tool should be justified, especially considering the availability of more updated alternatives.
- The limitations of the PREP-Mt method should be more extensively discussed.
- Phylogenetic Analysis:
- The biological significance of the obtained relationships should be more thoroughly discussed, including potential hybridization events (e.g., odorata vs M. foetida).
- The limitations of using only chloroplast DNA (absence of nuclear genome data) should be acknowledged.
- Figures and Tables:
- Some figures need more detailed legends, explaining abbreviations and symbols.
- On plots (e.g., Ka/Ks values), it would be beneficial to indicate thresholds (e.g., Ka/Ks = 1 line).
Questions for the Authors:
- What criteria were used in selecting the sampling locations?
- Are there additional data confirming the taxonomic identity of the studied individuals (e.g., voucher specimens)?
- Are there plans for further comparative studies on a larger number of wild Mangifera species?
- Is it possible to make the alignment files and phylogenetic trees used in the analyses publicly available?
- Do the authors intend to conduct functional validation of the genes identified under positive selection (e.g., petB, cemA)?
What Should Be Improved Before Publication:
- Professional English language editing.
- Full description of bioinformatics methods and sample selection criteria.
- Expanded positive selection analysis or clearly stating limitations.
- Provision of raw data accession numbers and links.
- Expanded interpretation of phylogenetic results, including potential hybridization scenarios.
The article is of high scientific value and certainly contributes new data to Mangifera chloroplast genomics research. However, I recommend acceptance only after major revisions — especially concerning language quality, methodology, and data presentation.
Author Response
Major Points for Revision:
- English Language Quality:
The text contains numerous grammatical, stylistic, and punctuation errors. Examples include:
- “and it was widely planted by the reason of high economic and nutrient value” — incorrect structure.
- Frequent repetitions, poor sentence construction, and tense inconsistencies.
âž” Suggestion: A full professional language editing is necessary.
Response: We sincerely appreciate the reviewer's careful reading and valuable suggestions regarding language quality. We acknowledge that the original manuscript contained some grammatical and stylistic issues, including the examples helpfully pointed out by the reviewer. In response, we have sent the manuscript for professional language editing by MDPI author services.
- Gaps in Methodology:
- Parameters used for data processing in Gblocks (e.g., block trimming settings) are not fully described.
Response: Thanks for the suggestion. We have reconducted phylogenetic tree and revised related Material and Method section in the revised version.
- The sample selection strategy (especially the two indica individuals from different countries) should be justified.
Response: We thank the reviewer for raising these important questions about sampling methodology. Samples were selected to maximize geographic representation across major distribution zones of M. indica and related species. The two M. indica samples were used for comparing from intra-species variation across the Asian cultivation. We have added the details in Introduction section.
- No detailed information is provided on sequencing depth (coverage), which is crucial for evaluating assembly quality.
Response: We appreciate the reviewer’s emphasis on sequencing depth for assembly validation. In our study, all samples were sequenced to a minimum coverage depth of 100×. We have added the sequencing depth in the revised manuscript.
- Positive Selection (Ka/Ks) Analysis:
- Merely stating Ka/Ks > 1 is insufficient to claim positive selection without appropriate statistical tests (e.g., FUBAR, MEME).
âž” Suggestion: Strengthen the selection analysis with statistical tests or explicitly discuss its limitations.
Response: We sincerely appreciate the reviewer's insightful suggestions regarding the rigor of positive selection analysis. To address this concern, we employed the branch model of PAML for Ka/Ks calculation, which utilizes a phylogenetic tree-guided multiple sequence alignment rather than pairwise comparisons. We have revised Figure 4 and expanded the Materials and Methods section to reflect these methodological updates.
- Raw Data Availability:
- Authors state that the data are publicly available, but no accession numbers for raw reads (e.g., in the SRA database) are provided.
âž” Suggestion: Add accession numbers and direct links.
Response: thanks for the suggestion. We have provided the raw data accession numbers in Data Availability Statement section.
- RNA Editing Analysis:
- The use of PREP-Mt as the predictive tool should be justified, especially considering the availability of more updated alternatives.
- The limitations of the PREP-Mt method should be more extensively discussed.
Response: thanks for the suggestions. We have claimed the limitations of PREP-Mt in the revised version (Line 263-271)
- Phylogenetic Analysis:
- The biological significance of the obtained relationships should be more thoroughly discussed, including potential hybridization events (e.g., odorata vs M. foetida).
- The limitations of using only chloroplast DNA (absence of nuclear genome data) should be acknowledged.
Response: thanks for the suggestion. We have thoroughly discussed the potential hybridization event and mentioned the limitations of using only cpDNA.
- Figures and Tables:
- Some figures need more detailed legends, explaining abbreviations and symbols.
Response: we have provided more detailed legends in the revised version.
- On plots (e.g., Ka/Ks values), it would be beneficial to indicate thresholds (e.g., Ka/Ks = 1 line).
Response: we have indicated threshold in Figure 4.
Questions for the Authors:
- What criteria were used in selecting the sampling locations?
Response: We thank the reviewer for raising these important questions about sampling methodology. Samples were selected to maximize geographic representation across major distribution zones of M. indica and related species. The two M. indica samples were used for comparing from intra-species variation across the Asian cultivation. We have added the details in Introduction section.
- Are there additional data confirming the taxonomic identity of the studied individuals (e.g., voucher specimens)?
Response: We are sorry that formal herbarium accession numbers are unavailable due to the non-institutional nature of some preserved samples. But collected materials were preserved using standardized field techniques. We have added this detail in Material and Method section.
- Are there plans for further comparative studies on a larger number of wild Mangifera species?
Response: We appreciate the reviewer's insightful suggestions regarding expanded taxonomic sampling. While beyond the current study's scope due to limited availability of authenticated wild material. We've added explicit discussion of these research priorities in future projects.
- Is it possible to make the alignment files and phylogenetic trees used in the analyses publicly available?
Response: Yes. All methodological details for phylogenetic tree construction are explicitly described in our manuscript. And raw data are publicly available under BioProject PRJNA1264534 (SRA: SRR33617711–SRR33617716).
- Do the authors intend to conduct functional validation of the genes identified under positive selection (e.g., petB, cemA)?
Response: We appreciate the reviewer's insightful suggestions regarding expanded functional validation. We fully agree these represent critical next steps. We've added this in result section of Ka/Ks and will pursue them in future projects.
What Should Be Improved Before Publication:
- Professional English language editing.
Response: We sincerely appreciate the reviewer's careful reading and valuable suggestions regarding language quality. We acknowledge that the original manuscript contained some grammatical and stylistic issues, including the examples helpfully pointed out by the reviewer. In response, we have sent the manuscript for professional language editing by MDPI author services.
- Full description of bioinformatics methods and sample selection criteria.
Response: Thanks for the suggestion. We have reconducted phylogenetic tree and revised related Material and Method section in the revised version.
- Expanded positive selection analysis or clearly stating limitations.
Response: we have expanded positive selection analysis.
- Provision of raw data accession numbers and links.
Response: thanks for the suggestion. We have provided the raw data accession numbers in Data Availability Statement section.
- Expanded interpretation of phylogenetic results, including potential hybridization scenarios.
Response: thanks for the suggestion. We have thoroughly discussed the potential hybridization event in the revised manuscript.
Reviewer 2 Report
Comments and Suggestions for Authors
In this manuscript, the authors present a comparative analysis of the complete chloroplast (cp) genomes of six Mangifera samples, including M. siamensis for the first time. The study aims to define the structure of the Mangifera cp genome, identify highly divergent regions suitable for DNA barcodes, and determine evolutionary relationships within the genus. The authors sequenced and analyzed the cp genomes, examining aspects such as genome structure, codon usage, RNA editing sites, repeat sequences, and phylogenetic relationships. The findings provide insights into the phylogenetic reconstruction and evolutionary processes within the Mangifera genus. Overall, the purposes and results are rigorous revisions based on the following comments.
Major comments
Many parts of the manuscript should be reformatted to meet the guidelines of Genes. For example, affiliation numbers should be correctly formatted, and the email addresses of all authors should be provided.
Please rewrite the abstract to be up to 200 words.
The abstract refers to five Mangifera species, while the materials and methods section details six samples. The results mention seven Mangifera specimens in the preliminary dataset. This inconsistency needs clarification to accurately represent the scope of the study.
The rationale behind choosing Citrus aurantifolia as the outgroup should be provided, along with a justification for using the TVM+F+I+G4 model for phylogenetic tree construction.
The quality of all phylogenetic trees is very poor. Please provide high-quality trees using maximum likelihood methods with at least 1000 bootstrap values.
The parameters and detailed methods for the bioinformatic analyses should be provided.
The origin and voucher information for each of the six Mangifera samples are not provided. This information is important for reproducibility and should be included in the manuscript.
The introduction mentions the need for comparative analyses between cp genomes, such as SSR, Ka/Ks, codon analysis, and RNA editing, but the results and discussion section does not delve deeply into these aspects. A more thorough exploration of these analyses is needed to support the study's claims.
Minor comments
Increase the size of Figure 1. It is too small to read.
Redraw Figure 2. I cannot see the numbers in the graphs. Magnify font sizes and use different colors according to the background.
The figure legends in most figures are poorly described.
I suggest regenerating graphs using Arial font.
Regarding Figure 4: Do the authors think it is informative or not? It is too small and contains too much information in one graph. Provide alternative ways to visualize the data efficiently.
Increase the size of Figure 6. Why are there seven Mangifera samples, not six?
Figures 7, 8, and 9 should be replaced with better quality images.
Throughout the manuscript, there are several typos and grammatical errors that need to be corrected for clarity and professionalism.
There is an incomplete sentence in the results section that needs to be revised for clarity and completeness.
The statement about the LSC regions exhibiting lower variability compared to the IR and SSC regions needs further clarification and context.
Ensure that all figures and tables are of high quality, easily readable, and appropriately labeled. Any abbreviations used should be defined in the figure captions.
Some references are missing from the reference list and should be added to ensure proper attribution and avoid plagiarism.
Line 3: "Yujuan Tang123#, Xiangyan Yang123#, Shixing Luo123*, Guodi Huang123*, Yu Zhang123, Ying Zhao123, Riwang Li123," - It is unconventional to have numbers as part of author names. Those numbers probably indicate affiliations and should be formatted accordingly.
Line 23: "The number of RNA-editing sites ranged from 60 to 61, and ndhB had the most RNA ed-iting sites in all species." - Awkward phrasing. Consider rephrasing for clarity, such as "The number of RNA editing sites ranged from 60 to 61, with ndhB exhibiting the highest number of RNA editing sites across all species."
Line 24: "Seven genes, atpB, cemA, clpP, ndhD, petB, petD and ycf15 exhibited Ka/Ks value >1, which would be suffered from the positive selection." - Grammatical errors. It should be "Seven genes, atpB, cemA, clpP, ndhD, petB, petD, and ycf15, exhibited a Ka/Ks value >1, suggesting they may be under positive selection." Also, gene names should be italicized.
Line 27: "Conclusions: Our comprehensive analysis of the whole cp genomes of the five Mangifera samples..." - The abstract refers to five species, while the methods section indicates six samples. This inconsistency needs to be resolved.
Line 35: "Genus Mangifera has 69 species of Mangifera around world that are mainly distrib-uted in tropical..." - Grammatical error. Consider: "The genus Mangifera comprises 69 species distributed around the world, mainly in tropical..." Also, "Mangifera" should be italicized.
Line 38: "...fruit. and it was widely..." - Remove the extra "and".
Line 39: "...by the reason of high economic..." - Should be "...because of its high economic...".
Line 50: "A key challenge stems from the predominantly outcrossing nature of both cultivated and wild mangoes, which are generally self-incompatible." - Awkward phrasing, consider: "A key challenge stems from the predominantly outcrossing nature of cultivated and wild mangoes, as they are generally self-incompatible."
Line 75: "Niu et al. have used whole cp genomes of five Mangifera species to investigate the evolutionary relationships within the genus." - It is not a complete sentence.
Line 83: "In this study, the cp genomes of 6 Mangifera samples were sequenced, with M. sia-mensis being sequenced for the first time." - awkward phrasing, consider: "In this study, we sequenced the cp genomes of six Mangifera samples, including M. siamensis for the first time." Mangifera and M. siamensis should be italicized.
Line 87: "...previously reported one (M. longipes)." - It should be "previously reported one (M. longipes)." M. longipes should be italicized.
Line 94: "The leaves of 6 samples belong to 5 species in genus Mangifera were collected in this study." - Awkward wording. Better: "We collected leaves from six samples representing five species of the genus Mangifera." The genus Mangifera should be italicized.
Line 115: "...PCGs in each of 7 samples by using Unipro ugenes v.36.0." - There is a space between "Unipro" and "ugenes."
Line 119: "...software to identified the RNA..." - Should be "...software to identify the RNA..."
Line 135: "Then we use IRscope to analysis IR Region Contraction (https://ir-scope.shinyapps.io/irapp/)." - Should be "Then, we used IRscope to analyze IR Region Contraction (https://ir-scope.shinyapps.io/irapp/)." Also, the sentence does not read very well.
Line 147: "The Fast The sequence of PCGs of Mangifera samples were extracted using Unipro ugenes" - Remove "The Fast".
Line 150: "The TVM+F+I+G4 model was used and the phylogenomic tree was constructed by IQtree 2.0 with 1000 bootstrap and maximum-likehood method." - "maximum-likehood" should be "maximum-likelihood".
Line 154: "The preliminary dataset of seven Mangifera specimens was meticulously filtered..." - Again, there's an inconsistency in the number of samples (five in the title, six in methods, seven here).
Line 169: "Previous research has indicated that the LSC regions exhibited lower variability compared to the IR and SSC regions." - This statement is then contradicted by the authors' findings. This either needs to be removed or explained more thoroughly.
Line 174: "We also detected around 133 to 134 genes in 6 species, they all had 37 tRNA. In additional M. indica TH had 89 protein-coding genes (PCGs) and others had 88 PCGs, so the total genes of M. indica TH was 134 and others were 133 (Table 1)." - Very confusing sentence. Rewrite for clarity.
Comments on the Quality of English LanguageThroughout the manuscript, there are several typos and grammatical errors that need to be corrected for clarity and professionalism.
Author Response
Major comments
Many parts of the manuscript should be reformatted to meet the guidelines of Genes. For example, affiliation numbers should be correctly formatted, and the email addresses of all authors should be provided.
Response: thanks for the suggestions. We have formatted the affiliation numbers and provided the email addresses of all authors.
Please rewrite the abstract to be up to 200 words.
Response: Thank you for your suggestion regarding the abstract length. We have carefully reviewed the journal's current template and confirmed that the abstract word limit is 250 words.
The abstract refers to five Mangifera species, while the materials and methods section details six samples. The results mention seven Mangifera specimens in the preliminary dataset. This inconsistency needs clarification to accurately represent the scope of the study.
Response: We sincerely appreciate the reviewer's careful attention to the numerical details of our study. The apparent inconsistencies arise from the following clarifications: The study sequenced 5 distinct Mangifera species, with M. indica represented by two geographically distinct samples (India and China) and the other 4 species each by one sample, totaling 6 samples in the Materials and Methods section. The Results section includes 7 specimens because we incorporated one published genome (M. longipes) for comparative evolutionary analysis, as noted in the revised text (Section 3.3). The revised manuscript includes expanded figure captions to eliminate ambiguity.
The rationale behind choosing Citrus aurantifolia as the outgroup should be provided, along with a justification for using the TVM+F+I+G4 model for phylogenetic tree construction.
The quality of all phylogenetic trees is very poor. Please provide high-quality trees using maximum likelihood methods with at least 1000 bootstrap values.
Response: Thanks for the suggestion. Citrus (Rutaceae) and Mangifera (Anacardiaceae) are phylogenetically distinct but share membership in the Sapindales order. We have added the reason in the revised version. In addition, we have reconducted phylogenetic tree.
The parameters and detailed methods for the bioinformatic analyses should be provided.
Response: we have revised related Material and Method section and provided details in the revised version.
The origin and voucher information for each of the six Mangifera samples are not provided. This information is important for reproducibility and should be included in the manuscript.
Response: We are sorry that formal herbarium accession numbers are unavailable due to the non-institutional nature of some preserved samples. But collected materials were planted and preserved using standardized field techniques. We have added this detail in Material and Method section.
The introduction mentions the need for comparative analyses between cp genomes, such as SSR, Ka/Ks, codon analysis, and RNA editing, but the results and discussion section does not delve deeply into these aspects. A more thorough exploration of these analyses is needed to support the study's claims.
Response: We have provided a more thorough exploration of these analyses in the revised version.
Minor comments
Increase the size of Figure 1. It is too small to read.
Response: we have increased the size of Figure 1.
Redraw Figure 2. I cannot see the numbers in the graphs. Magnify font sizes and use different colors according to the background.
The figure legends in most figures are poorly described.
I suggest regenerating graphs using Arial font.
Response: we have increased the number in graph and regenerated graphs using Arial font.
Regarding Figure 4: Do the authors think it is informative or not? It is too small and contains too much information in one graph. Provide alternative ways to visualize the data efficiently.
Response: we have redrawn Figure 4 to make it clear.
Increase the size of Figure 6. Why are there seven Mangifera samples, not six?
Response: We sincerely appreciate the reviewer's careful attention to the numerical details of our study. The Figure 6 includes 7 specimens because we incorporated one published genome (M. longipes) for comparative evolutionary analysis, as noted in the revised text (Section 3.3). The revised manuscript includes expanded figure captions to eliminate ambiguity.
Figures 7, 8, and 9 should be replaced with better quality images.
Response: Thanks for the suggestions. We have replaced Figure 7 and 8, and have redrawn Figure 9 with better quality images.
Throughout the manuscript, there are several typos and grammatical errors that need to be corrected for clarity and professionalism.
There is an incomplete sentence in the results section that needs to be revised for clarity and completeness.
Response: We sincerely appreciate the reviewer's careful reading and valuable suggestions regarding language quality. We acknowledge that the original manuscript contained some grammatical and stylistic issues, including the examples helpfully pointed out by the reviewer. In response, we have sent the manuscript for professional language editing by MDPI author services.
The statement about the LSC regions exhibiting lower variability compared to the IR and SSC regions needs further clarification and context.
Response: we have removed the sentence.
Ensure that all figures and tables are of high quality, easily readable, and appropriately labeled. Any abbreviations used should be defined in the figure captions.
Response: we have provided more detailed legends in the revised version.
Some references are missing from the reference list and should be added to ensure proper attribution and avoid plagiarism.
Response: the plagiarism was due to our preprint version online. We have explained the situation to editor.
Line 3: "Yujuan Tang123#, Xiangyan Yang123#, Shixing Luo123*, Guodi Huang123*, Yu Zhang123, Ying Zhao123, Riwang Li123," - It is unconventional to have numbers as part of author names. Those numbers probably indicate affiliations and should be formatted accordingly.
Response: yes, the numbers indicate affiliations and we have formatted in the revised version.
Line 23: "The number of RNA-editing sites ranged from 60 to 61, and ndhB had the most RNA ed-iting sites in all species." - Awkward phrasing. Consider rephrasing for clarity, such as "The number of RNA editing sites ranged from 60 to 61, with ndhB exhibiting the highest number of RNA editing sites across all species."
Response: thanks for the suggestion. We have rephrasing this sentence.
Line 24: "Seven genes, atpB, cemA, clpP, ndhD, petB, petD and ycf15 exhibited Ka/Ks value >1, which would be suffered from the positive selection." - Grammatical errors. It should be "Seven genes, atpB, cemA, clpP, ndhD, petB, petD, and ycf15, exhibited a Ka/Ks value >1, suggesting they may be under positive selection." Also, gene names should be italicized.
Response: thank you for the constructive feedback. We have revised the sentence accordingly.
Line 27: "Conclusions: Our comprehensive analysis of the whole cp genomes of the five Mangifera samples..." - The abstract refers to five species, while the methods section indicates six samples. This inconsistency needs to be resolved.
Response: we have corrected the abstract.
Line 35: "Genus Mangifera has 69 species of Mangifera around world that are mainly distrib-uted in tropical..." - Grammatical error. Consider: "The genus Mangifera comprises 69 species distributed around the world, mainly in tropical..." Also, "Mangifera" should be italicized.
Response: thank you for the suggestion. We have rewritten the sentence.
Line 38: "...fruit. and it was widely..." - Remove the extra "and".
Response: thanks for the suggestion. We have rephrasing this sentence.
Line 39: "...by the reason of high economic..." - Should be "...because of its high economic...".
Response: thanks for the suggestion. We have rephrasing this sentence.
Line 50: "A key challenge stems from the predominantly outcrossing nature of both cultivated and wild mangoes, which are generally self-incompatible." - Awkward phrasing, consider: "A key challenge stems from the predominantly outcrossing nature of cultivated and wild mangoes, as they are generally self-incompatible."
Response: thanks for the suggestion. We have rephrasing this sentence.
Line 75: "Niu et al. have used whole cp genomes of five Mangifera species to investigate the evolutionary relationships within the genus." - It is not a complete sentence.
Response: thanks for the suggestion. We have rephrasing this sentence.
Line 83: "In this study, the cp genomes of 6 Mangifera samples were sequenced, with M. sia-mensis being sequenced for the first time." - awkward phrasing, consider: "In this study, we sequenced the cp genomes of six Mangifera samples, including M. siamensis for the first time." Mangifera and M. siamensis should be italicized.
Response: thanks for the suggestion. We have rephrasing this sentence.
Line 87: "...previously reported one (M. longipes)." - It should be "previously reported one (M. longipes)." M. longipes should be italicized.
Response: thanks for the suggestion. We have rephrasing this sentence.
Line 94: "The leaves of 6 samples belong to 5 species in genus Mangifera were collected in this study." - Awkward wording. Better: "We collected leaves from six samples representing five species of the genus Mangifera." The genus Mangifera should be italicized.
Response: thanks for the suggestion. We have rephrasing this sentence.
Line 115: "...PCGs in each of 7 samples by using Unipro ugenes v.36.0." - There is a space between "Unipro" and "ugenes."
Response: thanks for the suggestion. We have rephrasing this sentence.
Line 119: "...software to identified the RNA..." - Should be "...software to identify the RNA..."
Line 135: "Then we use IRscope to analysis IR Region Contraction (https://ir-scope.shinyapps.io/irapp/)." - Should be "Then, we used IRscope to analyze IR Region Contraction (https://ir-scope.shinyapps.io/irapp/)." Also, the sentence does not read very well.
Response: thanks for the suggestion. We have rephrasing this sentence.
Line 147: "The Fast The sequence of PCGs of Mangifera samples were extracted using Unipro ugenes" - Remove "The Fast".
Response: thanks for the suggestion. We have rephrasing this sentence.
Line 150: "The TVM+F+I+G4 model was used and the phylogenomic tree was constructed by IQtree 2.0 with 1000 bootstrap and maximum-likehood method." - "maximum-likehood" should be "maximum-likelihood".
Response: thanks for the suggestion. We have reconducted the phylogenomic tree and rewritten this section.
Line 154: "The preliminary dataset of seven Mangifera specimens was meticulously filtered..." - Again, there's an inconsistency in the number of samples (five in the title, six in methods, seven here).
Line 169: "Previous research has indicated that the LSC regions exhibited lower variability compared to the IR and SSC regions." - This statement is then contradicted by the authors' findings. This either needs to be removed or explained more thoroughly.
Response: we have removed the sentence.
Line 174: "We also detected around 133 to 134 genes in 6 species, they all had 37 tRNA. In additional M. indica TH had 89 protein-coding genes (PCGs) and others had 88 PCGs, so the total genes of M. indica TH was 134 and others were 133 (Table 1)." - Very confusing sentence. Rewrite for clarity.
Response: thanks for the suggestion. We have rewritten this sentence.
Throughout the manuscript, there are several typos and grammatical errors that need to be corrected for clarity and professionalism.
Response: we have sent the manuscript for professional language editing by MDPI author services.
Round 2
Reviewer 1 Report
Comments and Suggestions for Authors
The authors have now submitted a revised version of the manuscript, along with a cover letter in which they have thoroughly and thoughtfully addressed the comments raised by the reviewers. Their responses are comprehensive and reflect a strong commitment to the revision process. They have introduced a number of significant changes, both in terms of content and the presentation of data, which have considerably improved the overall quality of the manuscript.
On the substantive level, the authors clarified methodological descriptions, added missing information, and expanded the discussion of the results, placing them more effectively within the context of current scientific knowledge. The phylogenetic interpretations have also been refined, resulting in a more coherent and convincing analysis. Editorially, they have enhanced the structure of the text, improved the language, and increased the clarity of tables and figures. All these changes have contributed to greater transparency and cohesion throughout the manuscript.
As a result of the revision, the article has gained in scientific value and now stands as a solid, well-documented study that represents a meaningful contribution to research on chloroplast genomes in Mangifera species. The revised manuscript not only meets the standards of the journal Genes, but is also likely to attract the interest of a broader community of researchers in plant genetics, systematics, and molecular evolution.
Reviewer 2 Report
Comments and Suggestions for Authors
The authors have diligently addressed the reviewers' comments.
The figures are now satisfactory, and many sections of the manuscript have been significantly improved.
I recommend the manuscript for publication in its current form.